# communications
## engineering

# Real-time multimodal sensory detection using widefield hippocampal calcium imaging

Dechuan Sun [1,2 ✉], Yang Yu[2], Forough Habibollahi[3], Ranjith Rajasekharan Unnithan[2] & Chris French [1 ✉]

The hippocampus is a complex structure that has a major role in learning and memory. It also integrates information from multisensory modalities, supporting a comprehensive cognitive map for both spatial and non-spatial information. Previous studies have been limited to real-time spatial decoding, typically using electrodes. However, decoding hippocampal non-spatial information in real time has not been previously described. Here, we have constructed a real-time optical decoder driven by the calcium activity of large neuronal ensembles to decode spatial, visual, and auditory information effectively. Using advanced machine learning techniques, our rapid end-to-end decoding achieves high accuracy and provides a multisensory modality detection method. This method enables the real-time investigation of hippocampal neural coding and allows for direct neural communication with animals and patients affected by functional impairments. The ability to decode multimodal sensory inputs in real time thus forms the basis for an all-optical brain-computer interface.

[1] Neural Dynamics Laboratory, Department of Medicine, The University of Melbourne, Melbourne, Victoria, Australia. [2] Department of Electrical and Electronic Engineering, The University of Melbourne, Melbourne, Victoria, Australia. [3] Department of Biomedical Engineering, The University of Melbourne, Melbourne, Victoria, Australia. ✉email: dechuan.sun@unimelb.edu.au; frenchc@unimelb.edu.au

The hippocampus, embedded deep within the temporal cortices of humans and animals, connects with many other brain structures directly or indirectly and plays an important role in memory and cognition. It has been demonstrated to support a spatial cognitive map, providing an environment-centric spatial memory system[1,2]. Apart from encoding spatial information, previous studies have revealed that the hippocampal neuronal network also encodes non-spatial information such as visual[3,4], auditory[5–7], olfactory[8,9], gustatory[10], and tactile[11,12]. These observations indicate a more abstract and comprehensive hippocampal cognitive map, formally generating a high-dimensional space for both spatial and non-spatial information. Accurately decoding this mapping in real time would facilitate direct neural communication in both experimental and potentially clinical scenarios, providing an effective communication channel as a brain-computer interface. However, these non-spatial responses have been found to be spatial context-dependent or task-dependent in almost all previous studies (see O'Keefe and Krupic for review)[13]. As a result, it is plausible that such responses may actually reflect the specific location or context in which a particular feature is present rather than the feature itself. It remains unclear whether non-spatial responses can be consistently decoded without concurrent spatial input, especially in real-time scenarios.

To date, only hippocampal spatial information has been resolved in real time using electrophysiological signals[14–17] or calcium signals[18]. Detecting hippocampal non-spatial information in real time is complex and challenging: several recording channels are needed, the sensitivity of hippocampal neurons to non-spatial information is low, analysis pipelines are time-consuming, electrodes often shift, and signals tend to attenuate over time.

Here, we designed an optical brain-computer interface (OBCI, Fig. 1a) based on a single-photon imaging technique (miniscope)[19] in animals traversing a linear track and exposed to light stimuli or tones centered at three different frequencies. While both Tu et al.[18] and our study used different decoders for position reconstruction, a noteworthy innovation in our work is the integration of a Kalman filter after neural network decoding to reduce inherent decoding noise. Importantly, experiments involving non-spatial stimuli were conducted within a small recording chamber to eliminate spatial inputs. Using raw neuronal calcium activity instead of deconvoluted calcium events, and machine learning models, we were able to reconstruct the animal's running trajectory and identify visual and auditory stimuli in real time (Fig. 1b–d). We achieved low decoding errors with 9 cm/frame in position reconstruction, 3% in visual stimuli identification, and 17% in auditory stimuli identification. This study presents a demonstration of hippocampal multisensory modality decoding using a multimodal real-time OBCI.

## Results

In our experiments, we utilized a frame rate of 30 Hz, resulting in a frame interval of approximately 33 ms. To ensure consistent frame-by-frame decoding, it was crucial to maintain data processing times below 33 ms to prevent camera data from overflowing the PC buffer. The footprints of neurons are detected offline, and the extraction of raw calcium signals only requires simple matrix operation. The total processing time, which included image registration and calcium signal extraction, was approximately 2.2 ms. This efficient processing time rendered our system suitable for real-time decoding.

**Position reconstruction.** Each mouse ($n = 3$) first underwent a training session in a linear track to construct a position

reconstruction model (Fig. 1). During the training sessions, we identified a large population of hippocampal neurons in each mouse ($n = 781, 478, 622$). We measured the spatial information content of each neuron and detected many position-sensitive neurons ($n = 322, 320, 250$). An example of the place field map of sensitive neurons is shown in Fig. 2a. The raw fluorescent intensities of these neurons were used to train the position reconstruction model. Using the four models with a Kalman filter, we could accurately reconstruct the running trajectories of the mice. The Gaussian naïve Bayes (GNB) decoder showed the highest decoding error (mean: 22.79 ± 3.42 cm/frame; median: 16.00 ± 2.64 cm/frame), while the other three decoders achieved better performance with comparable decoding errors (mean: ~14 cm/frame; median: ~11 cm/frame; Fig. 2b, c, Supplementary Table 1). An example of the position reconstruction using the training data is shown in Fig. 2d. Intriguingly, we observed various firing patterns of sensitive neurons, indicating that precise position reconstruction required more subtle features, which the GNB decoder was incapable of detecting (Supplementary Fig. S1a, b).

In the real-time session, the raw fluorescent intensities of location-sensitive neurons detected in the training session were extracted and fed into the decoding model. Additionally, we generated a place field map of the location-sensitive neurons in the real-time session, which revealed distinct and specific firing locations for all neurons, as demonstrated in Fig. 2e. Our decoding model produced an impressive decoding accuracy, with a low average decoding error of 13.65 ± 0.50 cm/frame (13.20, 13.09, and 14.65 cm/frame, respectively) and a low median error of 9.33 ± 0.67 cm/frame (8.00, 10.00, and 10.00 cm/frame). An example of the trajectory reconstruction is shown in Fig. 2f, g and Supplementary Fig. S1c–e.

We then examined the noise levels of signals in both the training session and real-time session, but no significant differences were detected (Supplementary Fig. S2a, b). Moreover, the cross-correlation of place field maps between the two sessions indicated a peak correlation value at the origin of the coordinate (Supplementary Fig. S2c), validating the stability of neuronal firing patterns across sessions. We further evaluated the impact of long short-term memory (LSTM) window sizes on the decoding accuracy and discovered that a 5-frame window size (~0.17 s) provided accurate decoding, whereas a 20-frame window size resulted in substantially poorer performance (Supplementary Fig. S2d). We also analyzed the effect of neuron numbers on decoding accuracy in the presence of between 20 and 100% of sensitive neurons. We observed that using more neurons in the position reconstruction decreased decoding error (Supplementary Fig. S2e) by 15.68% over this range. Finally, we measured the frame processing time for different models and quantities of neurons. With 100% of sensitive neurons employed, the support vector machine (SVM) model was the slowest to process each frame data (~5 ms) compared with the other models that were much faster (~0.2 ms; Supplementary Fig. S2f).

**Visual stimuli identification.** We next studied whether the hippocampal ensemble activity could be decoded to identify visual inputs in each frame (Fig. 1). To achieve this, we individually tested three mice, each of which underwent a training session to develop the decoding model. In the field of view, we observed 536, 707, and 569 neurons in the three animals, respectively. The neuronal fluorescent intensity observed in light and dark environments showed similar distributions (Supplementary Fig. S3a). We measured the information content for each neuron and detected 204, 392, and 407 sensitive neurons, respectively. We next tested the performance of a GNB decoder, an SVM decoder, and a multilayer perceptron (MLP) neural

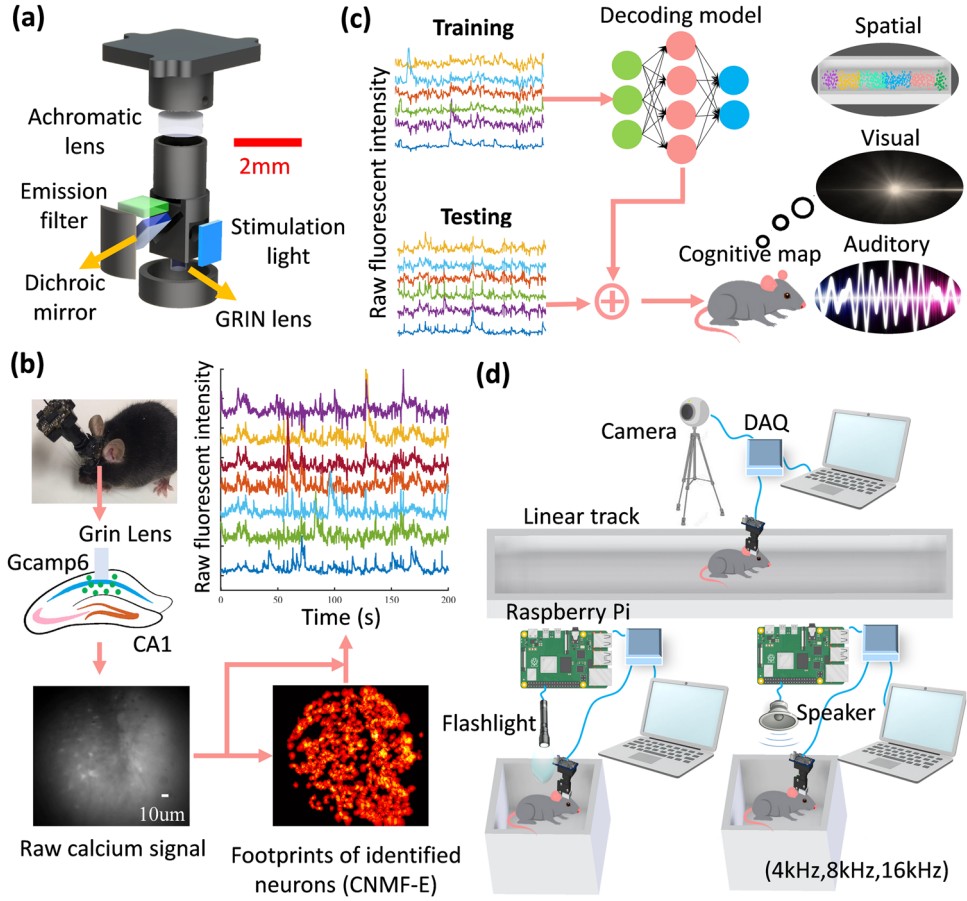

**Fig. 1 In vivo calcium signal recording and analysis pipelines. a** Mechanical and optical assembly of the miniscope. **b** A miniscope was used to image the activity of Gcamp6 labeled hippocampal neurons through a Grin lens. Using a constrained nonnegative matrix factorization algorithm for endoscopic recordings (CNMF-E) to identify the spatial footprints of neuronal populations, we extracted the raw calcium activity of detected neurons. **c** A miniscope and a data acquisition board (DAQ) were used to record the hippocampal activity in a mouse. In the position reconstruction experiment, a camera was used to track the location of a mouse traversing a 1.6 m linear track. In the visual/auditory identification experiment, a Raspberry Pi was used to turn a flashlight/speaker on and off to provide visual/auditory stimuli (sinusoidal tones at 4, 8, or 16 kHz) for a mouse within a small recording chamber. **d** Hippocampal activity decoding pipelines. In the training session, the raw fluorescent intensity of sensitive neurons was used to construct the decoding model. This model was then deployed in the real-time session to decode the spatial, visual, and auditory information.

network to identify the visual inputs based on the training data. Supplementary Table 2 summarizes the decoding accuracy of each model. In comparison to the GNB decoder (mean error: 20.54% ± 8.70%), the SVM decoder and MLP model both showed very high decoding accuracies (mean error: ~3%). This could be partially explained by observing features derived from the brain activity using the MLP model, which were quite distinguishable under two circumstances (Supplementary Fig. S3b, c). An example of the performance of different models is shown in Fig. 3a. We selected the model with the best performance to apply to each mouse in the subsequent real-time session.

In the real-time session, we used the raw calcium signals of sensitive neurons to predict the visual inputs. Notably, decoding errors were remarkably low in all three mice, with an overall average of 3.36% ± 1.47% (5.47%, 4.07%, and 0.53% in the respective mice). The decoding performance is illustrated in Fig. 3b and Supplementary Fig. S3d, e. Additionally, the average neuronal activity during light and dark epochs revealed distinct temporal firing patterns (Fig. 3c, d).

The noise levels of the signals in the training sessions and the real-time sessions did not show significant differences (Supplementary Fig. S4a, b). Intriguingly, the maximum cross-correlation value of firing rate maps between two sessions was not at the origin of the coordinate axes, indicating that neuronal

temporal firing patterns were not highly consistent across sessions (Supplementary Fig. S4c). This possibly implies that visual stimuli not only have temporal effects on neuronal activity but also affect the functional connectivity of neuronal populations. On the other hand, this could be attributed to sensory adaptation. The cross-correlation measured the similarity of the neuronal firing rate maps between the training session and the real-time session. It is important to note that this method primarily captures the fundamental characteristics of population activity and cannot discern deep features within the neuronal firing pattern. Considering that the neural network decoding model was able to provide accurate decoding even in this scenario, it may imply that neuronal activity related to non-spatial stimuli is better characterized by deeper features. This could potentially explain why some previous studies have reported a low sensitivity of hippocampal neurons to these non-spatial stimuli[13], possibly due to the analysis method. Increasing the number of neurons used in the decoding process resulted in higher decoding accuracy, as observed in position reconstruction experiments (Supplementary Fig. S4d). As the decoding model was simpler than that used in position reconstruction experiments, the MLP model only required about 0.01 ms to process the data in each frame (Supplementary Fig. S4e).

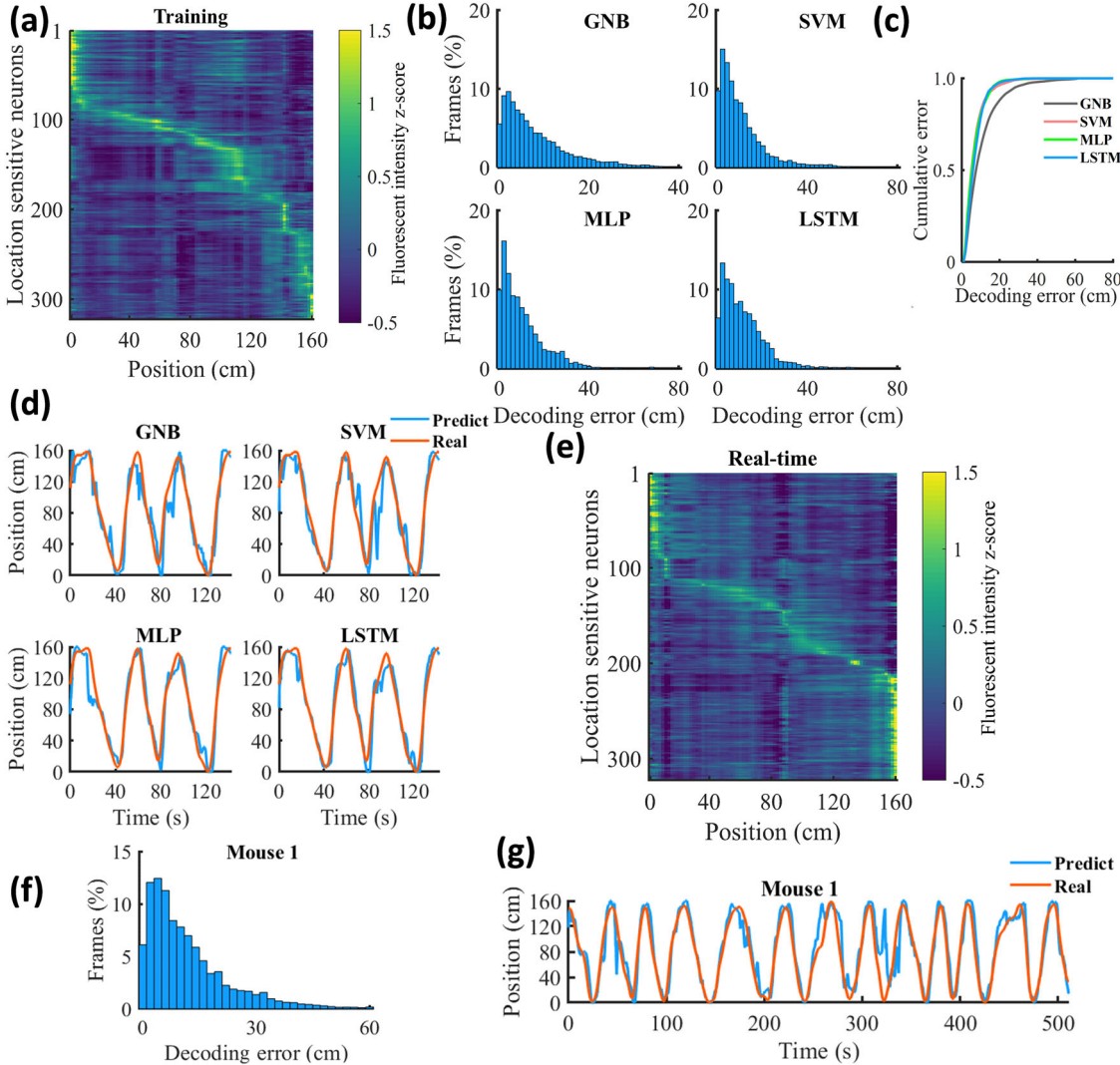

**Fig. 2 Position reconstruction experiment. a** An example of the place field map of sensitive neurons in training sessions. The color represents the fluorescent intensity z-score. **b** Decoding error histograms of five-fold cross-validation results in the training session using Gaussian naïve Bayes (GNB), support vector machine (SVM), multilayer perceptron (MLP), and long short-term memory (LSTM) decoders. **c** Cumulative fraction of the decoding error using different models (One-way analysis of variance, F(3,240) = 45.02, p-value < 0.0001; Bonferroni post-hoc test showed that decoding error using GNB decoder was significantly higher than that using SVM, MLP, and LSTM, p-value < 0.0001). **d** Decoding performance using different models in the training session. The decoding model was constructed based on the first 75% of the recorded data and checked on the remaining 25%. The red curve demonstrates the mouse's real position tracked by the video camera, and the blue curve represents the reconstructed position. **e** The place field map of sensitive neurons in the real-time session. **f** The decoding error histogram in the real-time position reconstruction experiment. **g** An example of the mouse's running trajectory reconstruction in a real-time session.

**Auditory stimuli identification.** Three mice were exposed to pure sinusoidal tones at frequencies of 4, 8, and 16 kHz to investigate if the activity of hippocampal neurons could be decoded to differentiate between the auditory stimuli frequencies (Fig. 1). We observed 534, 619, and 599 neurons in each mouse. Initially, we used a GNB decoder, an SVM decoder, and an MLP neural network model to identify the auditory input in each frame during the training session. However, none of these models were able to make accurate predictions. Supplementary Table 3 and Fig. 4a summarize the decoding error of each model. We next sectioned the data into epochs according to the frequency of stimuli, where each epoch contained a 2 s sound-on period followed by a 3 s sound-off period. The neuronal fluorescent intensity in different frequency epochs displayed similar distributions (Supplementary Fig. S5a), and we detected 198, 227, and 230 sensitive neurons in each mouse. An example of the

average neuronal activity of sensitive neurons is shown in Fig. 4b. GNB, SVM, and MLP models are less effective in handling temporal multi-dimensional data. A common approach to analyze such data is to flatten or reshape the inputs. However, this method significantly increases dimensionality, which brings about challenges such as the curse of dimensionality, heightened computational complexity, and the potential loss of spatial information in the features. Given these challenges, along with substantial optimization time consumption, we did not use these models for further analysis. We then trained a convolutional neural network (CNN) model to identify the frequency of each epoch and achieved high decoding accuracy with an error rate of 17.67%, 22.76%, and 22.89% in each mouse (mean ± SEM: 21.11% ± 1.48%). An example of CNN performance is shown in Fig. 4c, and features extracted from the model exhibit complex properties (Supplementary Fig. S5b, c).

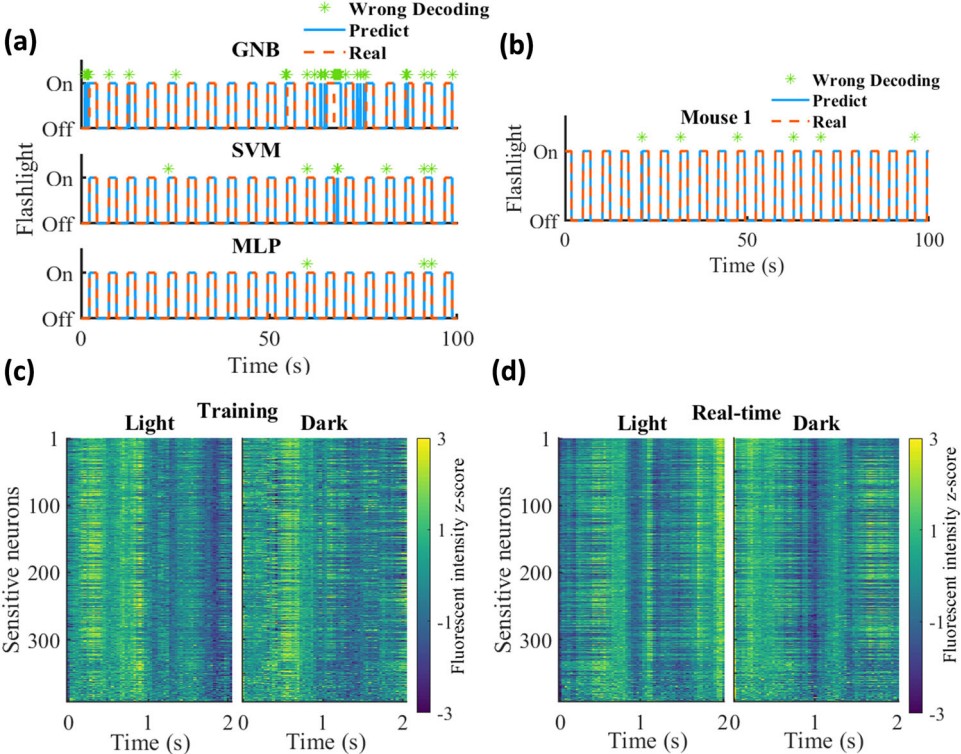

**Fig. 3 Visual stimuli experiment. a** An example of the decoding performance of different decoders (GNB Gaussian naïve Bayes, SVM support vector machine, MLP multilayer perceptron) in the training session (100 × 30 frames). The decoding model was constructed based on the first 75% of data and checked on the remaining 25% of data. The red dashed line and solid blue line represent the real and predicted statuses of the flashlight, respectively. The wrong decoding frame is marked with a green star. **b** An example of the decoding performance in the real-time session (100 × 30 frames). **c** The visual stimuli evoked neuronal activity of sensitive neurons in light and dark epochs in the training session. The color represents the fluorescent intensity z-score. The flashlight was turned on or off at time "0". **d** The visual stimuli evoked neuronal activity of sensitive neurons in light and dark epochs in the real-time session.

In the real-time session, the CNN model constructed in the training session was deployed to decode the raw calcium activity of sound-sensitive neurons. The resulting average temporal firing patterns exhibited similarity to those observed in the training session (Fig. 4d). For each mouse, the decoding error ratio was 13.25%, 20.78%, and 19.48% respectively (mean ± SEM: 17.83% ± 2.32%). An example of the decoding performance is shown in Fig. 4e and Supplementary Fig. S5d, e.

In the study, noise levels in the signals of both the training and real-time sessions were found to be similar, as indicated by the results in Supplementary Fig. S6a, b. Cross-correlation analysis of firing rate maps between two sessions revealed multiple peaks, with a dominant peak located at the origin, suggesting relatively consistent temporal firing patterns across sessions (Supplementary Fig. S6c). Likewise, the highest decoding accuracy was achieved using all sensitive neurons (Supplementary Fig. S6d) and the model consumed a short period of ~1.6 ms to process the 5 s epoch data using the CNN model (Supplementary Fig. S6e).

## Discussion
We have decoded multiple sensory modalities from hippocampal neuronal activity in mice, and our proposed method can recognize certain pre-trained patterns that are associated with specific behaviors. Our low-latency end-to-end analysis pipeline provided accurate decoding of both hippocampal spatial and non-spatial information, which allowed a real-time readout of these internal cognitive states, delivering a functional cognitive interface. The hippocampus supports a diverse cognitive map that incorporates both spatial and non-spatial information and therefore represents

a useful target for decoding multimodal information, which is particularly suited for multisensory decoding compared with other primary sensory brain regions.

The projection pathways of sensory information to the hippocampus differ anatomically. Spatial information mainly targets the dorsal and posterior hippocampus[20]. Guger et al., Sodkomkham et al., and Hu et al. have reconstructed rat running trajectories in real time using small sets of hippocampal neurons recorded electrophysiologically[14,15,17]. Non-spatial information has been reported to largely flow into the ventral and anterior hippocampus[20]. Visual signals project to the hippocampus from the visual cortex through a multi-synaptic pathway[21,22]. Previous studies found that hippocampal activity could represent visual stimuli[4] and neurons in the hippocampus and visual cortex showed synchrony in visual stimuli experiments[23]. The transmission pathways between the hippocampus and auditory cortex are complex. It is believed that there are two major pathways—the lemniscal pathway and the non-lemniscal pathway, and the auditory signaling to the hippocampus has likely undergone several integrative stages[7,24]. Neurons in the hippocampus of rodents have been reported to show a certain degree of sound sensitivity[6]. All these non-spatial sensitivity investigations made use of low channel counts electrophysiological recording methods but did not show whether the activity could be decoded in real time. In comparison, our optical BCI enabled the recording of larger neuronal populations, and we demonstrated that these non-spatial modalities could be detected in real time with high fidelity.

**Decoding models.** We tested different machine learning models to decode the hippocampal cognitive map, including the GNB

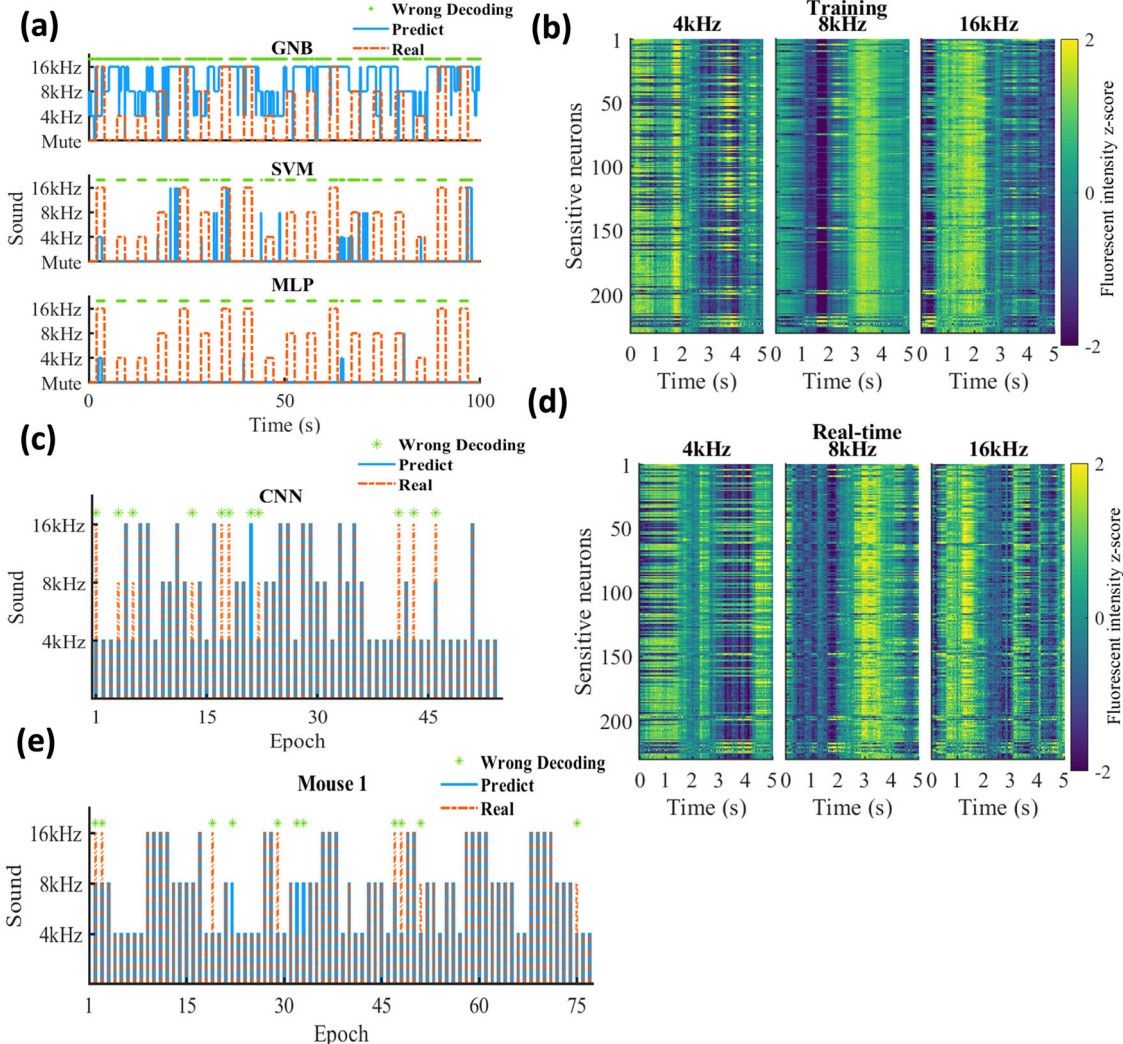

**Fig. 4 Auditory stimuli experiment.** The mice were exposed to sound stimuli at three different frequencies. **a** An example of the decoding performance of different models (GNB Gaussian naïve Bayes, SVM support vector machine, MLP multilayer perceptron) in the training session (100 × 30 frames). The decoding model was constructed based on the first 75% of data and checked on the remaining 25% of data. The red dashed line and solid blue line represent the real and predicted status of the auditory stimuli, respectively. The wrong decoding frame is marked with a green star. All decoders failed to make accurate predictions. **b** The sound stimuli evoked neuronal activity of sensitive neurons in 4, 8, and 16 kHz epochs during the training session. The color represents the fluorescent intensity *z*-score. The speaker was turned on at the time "0" and turned off at the time "2". **c** An example of the convolutional neural network (CNN) decoding performance in the training session. The data was sectioned into 5-s time-length epochs according to the frequency of the stimuli. **d** The sound stimuli evoked neuronal activity of sensitive neurons in the real-time session. **e** An example of the decoding performance using a CNN model in the real-time session.

decoder, SVM decoder, MLP neural network model, LSTM neural network model, and CNN model. In all cases, the GNB decoder showed the highest decoding error, which might be due to the nonlinearity of the hippocampal neuronal network and the inherently complex signal composition. Alternatively, the determination of adequate priors may have been suboptimal.

In both the position reconstruction and visual stimuli identification experiments, the MLP neural network model and SVM decoder showed similar decoding performance in all animals. Compared with the MLP neural network model, the SVM decoder has the advantage of optimizing fewer hyperparameters and can be trained in an online mode[25,26], which avoids a separate training session. The LSTM algorithm is an artificial recurrent neural network that can achieve good prediction accuracy from time-series data[27,28]. It simulates a biologically relevant model of neuronal activity processing. However, it did not show the best performance in position reconstruction

experiments, which was somewhat surprising. A possible explanation may be the slow kinetics of neuronal calcium activity. Action potentials cause calcium influx and efflux in excitable cell bodies, and depolarization-evoked neuronal firing has a long-lasting effect on calcium activity. This indicates that the calcium activity in the current frame inherits partial information encoded in previous frames, which might weaken the strength of memory units in the model. It is known that incorporating too much old information in an LSTM neural network model causes a drop in its performance because that information may not be relevant or useful or may introduce unwanted noise[29].

We showed that spatial and visual information could be decoded accurately in each recorded frame of activity. However, a relatively long time window was needed to decode the auditory information. This may be due to various encoding mechanisms for different types of sensory information. Indeed, auditory-

evoked neuronal activation has been reported to exhibit variable latencies[6,30], resulting in long-lasting temporal firing patterns that would be consistent with this hypothesis. Another explanation may be the relatively low absolute sensitivity of CA1 neurons to auditory stimuli[5,6], necessitating the highly optimized analysis and longer recording periods.

**Optical brain-computer interfaces (OBCIs).** Conventional intracranial brain-computer interfaces (BCIs) use electrode arrays to record neuronal action potentials or local field potentials to detect internal states. More recently, the potential for calcium imaging using multiphoton imaging of neuronal ensembles as a BCI has been examined[31,32]. The relationship between neuron action potentials and calcium activity is complex. An action potential activates voltage-gated calcium channels, eliciting a nonlinear rise in intracellular calcium concentration. Although dynamics of calcium activity are relatively slow, it has been shown to track action potential frequency[33]. In our experiments, we used GCaMP6f with relatively fast kinetics (~50 ms temporal resolution)[34]. Thus, the temporal resolution of calcium signals seems functionally comparable to electrical signals.

Clancy et al. utilized volitional control of a small number of neurons identified with two-photon imaging that could indirectly modulate the sound from a speaker[31]. Trautmann et al. implemented a two-photon OBCI in a head-fixed macaque for the detection of the animal's arm motion[32]. In contrast, our technique uses single-photon imaging, which yields poorer signal quality compared to two-photon imaging since it also collects neuronal firing activity of out-of-focus cells as well. However, it is important to note that this additional neuronal firing information has the potential to enhance decoding accuracy. Additionally, this workflow is low-cost and does not require bulky equipment. Interestingly, Chen et al. recently implemented an approach utilizing hippocampal calcium signals and simple linear decoders to estimate the positional information of Long-Evans rats within a linear track[35]. Despite the generally higher spatial specificity of rat place cells in comparison to mouse, the researchers reported a decoding error of 30 cm/frame, which is higher than the 9 cm/frame error observed in our experiments. Additionally, the authors implemented a relatively large spatial bin size of 20 cm in their analysis. Tu et al. reconstructed an animal's position in a sensory-cued treadmill using single-photon calcium imaging datasets[18]. Their maximum likelihood estimation method demonstrated good decoding accuracy, achieving a decoding error of around 6 cm. However, it is not clear whether this measurement refers to 6 cm per second or 6 cm per frame. Comparing decoding accuracy between their study and ours presents challenges due to differences in experimental design. These distinctions encompass varying recording sampling frequencies, which are known to impact calcium signal quality, as well as differences in experimental apparatus. It is important to note that the animal had to maintain a stable running speed on the treadmill, and running speed can impact place cell activity[36]. Furthermore, discrepancies in our training methods further complicate the comparison, as it is difficult to ascertain whether the animals were trained to the same level.

In the experiment, we used a relatively short 1.6 m linear track. This limitation is attributed to the confined space within the experimental room, with the issue of cable handling posing a more substantial challenge. Managing longer cables can become challenging as the linear track extends. However, one solution to address the cable-related challenges is to use a wire-free miniscope[37], which has recently become available on the market.

In summary, we have successfully decoded spatial, visual, and auditory information from the mouse hippocampus using widefield single-photon imaging in real time, constituting an effective OBCI. The end-to-end OBCI system proposed here presents a proof of concept for decoding the hippocampal cognitive map in real time. It expands the method and opportunity to study the activity of hippocampal neuronal ensembles and will be helpful for future content-specific closed-loop BCI experiments. Furthermore, it provides an approach for multisensory modality decoding, which may be applied in clinical applications and scientific research in the future. Additionally, future studies exploring the application of OBCI to decode action behaviors hold particular promise, offering potential benefits to patients affected by functional impairments.

## Methods

**Miniscope system design.** Our miniscope optical system was developed based on the previous design from UCLA[19]. The UCLA design is a compact fluorescence microscope consisting of a stimulation light source, half ball lens, excitation filter, dichroic mirror, GRIN lens, emission filter, and achromatic lens. We made changes to this system by replacing the original achromatic lens with a shorter focal length of 7.5 mm (45407, Edmund Optics) and the GRIN lens with a pitch value of 0.23 (64520, Edmund Optics). This modified optical system has a shorter focal length, lower magnification, and larger field of view, which are desirable features for BCI applications (Fig. 1a; Supplementary Fig. S7). A shorter focal length in the optical system can decrease the overall height of the miniscope. Lower magnification can reduce the requirement for utilizing all available CMOS sensor pixels, enabling more efficient real-time processing. A larger field of view can provide access to more neurons, improving decoding accuracy. In our practical implementation, it was observed that the inclusion of an excitation filter and half ball lens was not imperative for capturing high-quality images. The stimulation LED exhibits excellent monochromaticity and collimation, and the dichroic mirror additionally enhances the purity of the stimulation light. Removing these optics further reduces the size and weight of the miniscope. Supplementary Fig. S7d depicts a comparison of the imaging quality obtained with and without an excitation filter and a half ball lens.

**Animals and surgery.** Two stereotaxic surgeries were conducted on mice under anesthesia (isoflurane: 3%–5% induction, 1.5% maintenance). To label hippocampal neurons, mice were unilaterally injected with 500 nl of pAAV.Syn.GCaMP6f.W-PRE.SV40 virus (#100837-AAV1, AddGene) in the dorsal CA1 (right brain hemisphere, 2.1 mm posterior to the bregma, 2.1 mm lateral to the midline, and 1.65 mm ventral from the surface of the skull). One week following the injection, two anchor screws were secured into their skulls, and a circular craniotomy, 2 mm in diameter, was performed next to the injection site (2.1 mm posterior to the bregma, 1.6 mm lateral to the midline). The cortex above the corpus callosum was aspirated using a 27-gauge blunt needle attached to a vacuum pump, and a 1.8 mm diameter GRIN lens was implanted at a depth of 1.35 mm from the surface of the skull. The lens was fixed in place with cyanoacrylate glue and dental acrylic and protected with silicone adhesive (Dragon Skin® Series). Mice were given analgesics (carprofen:5 mg/kg; dexamethasone: 0.6 mg/kg) and enrofloxacin water (1:150 dilution, Baytril®) to recover for seven days. Neuronal calcium activity was measured 4–5 weeks later. After finding the best field of view, a baseplate was cemented on the head and a plastic cap was locked into the baseplate to protect the lens[38] (Fig. 1a).

**Information content analysis and neuronal sensitivity identification.** Information content can be used to quantify the

precision of neuronal coding with a large value indicating more precise coding. In the position reconstruction experiment, the definition of the information content is similar to that described previously[39,40], but we changed the measurement of neuronal firing rate to neuronal fluorescent intensity to adapt to our recording technique,

$$I = \frac{r_i}{\bar{r}} \log_2 \frac{r_i}{\bar{r}}, \bar{r} = \sum_{i=1}^{k} p_i r_i, \tag{1}$$

where $I$ represents the information content, $i$ is the spatial bin index (the linear track is divided into several 2 cm bins), $k$ is the number of spatial bins, $p_i$ is the probability of occupancy of the $i^{th}$ bin, $r_i$ is the average fluorescent intensity in the $i^{th}$ bin and $\bar{r}$ is the overall mean fluorescent intensity.

To identify neurons responsive to place, the animal's location was shuffled, and the information content was recomputed for each shuffle. This step was repeated 1000 times, and a neuron was marked as a location-sensitive neuron if its information content in the unshuffled trial exceeded 95% of the shuffled trials.

The definition of the information content for visual and auditory stimuli was the same, but the data binning was implemented in the temporal domain. There were two states (light or dark) in the visual experiments and three states denoted by three different frequencies in the auditory experiments. The same shuffling method was used to detect light and sound-sensitive neurons.

**Calcium activity decoder**. The performance of several models to decode neural signals was evaluated and compared. These included a Gaussian naïve Bayes (GNB) decoder, a support vector machine (SVM) decoder, a multilayer perceptron (MLP) neural network model, a long short-term memory (LSTM) neural network model, and a convolutional neural network (CNN) model. The decoders were constructed and implemented on the Python platform using the scikit-learn tool kit and TensorFlow library[41,42].

The GNB decoder is a type of probabilistic-based prediction algorithm based on Bayes' theorem. In the position reconstruction experiment, raw fluorescent intensities of location-sensitive neurons were first normalized to remove the mean and scaled to unit variance. Given a sequence of fluorescent intensities from location-sensitive neurons in each frame, the estimated position $\hat{y}$ was defined as,

$$\hat{y} = \arg\max_{y} P(y) \prod_{i=1}^{N} P(x_i|y) \tag{2}$$

$$P(x_i|y) = \frac{1}{\sqrt[2]{2\pi\sigma_y^2}} e^{-\frac{(x_i-\mu_y)^2}{2\sigma_y^2}} \tag{3}$$

where $P(y)$ is the probability of occupancy of bin $y$, $x_i$ is the normalized fluorescent intensity of the $i^{th}$ neuron, and $\mu_y$ and $\sigma_y$ represent the mean and standard deviation of the normalized fluorescent intensity of the $i^{th}$ neuron at bin $y$, respectively. In visual stimuli experiments, $y$ represents either light-bin or dark-bin. In auditory stimuli experiments, $y$ represents the bins with different frequencies.

The SVM decoder performs classifications by constructing a set of hyperplanes that maximizes the margins between different classes. The SVM model was trained and constructed using the normalized input data with a nonlinear radial basis function kernel. The kernel width, gamma, was set to be the reciprocal of the number of input features. A cost parameter "C" was

optimized by applying a grid search technique with five-fold cross-validation.

An MLP is an artificial neural network that is commonly used for solving prediction and classification problems, especially when the input data is not linearly separable. We built an MLP model with one input layer receiving normalized data, two hidden layers activated by a rectified linear unit (ReLU) function, and one output layer with softmax activation. An Adam optimizer and a categorical cross-entropy loss function were used to compile the model. The batch size was set to 32, and all the other hyperparameters, including the number of nodes in hidden layers, learning rate, and the number of epochs, were optimized by implementing a grid search method using five-fold cross-validation.

An LSTM model is a type of recurrent neural network that can use internal memory to process sequences of data with variable lengths and is extremely useful for time-series forecasting. The LSTM model was implemented to reconstruct the animal's running trajectory in our experiments. The model consisted of one input layer, two fully connected LSTM layers activated by a ReLU function, a dropout layer (dropout rate: 0.2) that prevented overfitting, and one output layer with a softmax activation function. In position reconstruction experiments, the output of the model was marked as the animal's current location bin and different lengths of normalized time-series data were evaluated to build and train the model. The same method as the MLP model was used to compile and optimize the model.

A CNN is an artificial neural network that has been frequently implemented in image processing but also shows good performance for time-series data. It is designed to detect spatial hierarchies of features in the input data. The CNN model was tested to decode the hippocampal activity in auditory stimuli experiments. The inputs were the time series of raw fluorescent intensities from sensitive neurons. The model contained two convolutional layers using ReLU activations, with a max-pooling layer added after each convolutional layer for dimensionality reduction. A dropout layer (dropout rate: 0.2) was then concatenated to prevent overfitting. Finally, a fully connected layer with a softmax activation function was added to output the probability distribution for each class. A grid search method was performed to determine the hyperparameters that yielded the highest decoding accuracy.

In our decoding methods, such as the GNB decoder, SVM decoder, and MLP neural network decoder, we used single-frame data as the input. However, when employing the LSTM neural network model, we tested different window lengths and determined that a temporal window of 5 frames typically provided accurate decoding (Supplementary Fig. S2d). As the auditory stimuli have a long temporal effect on neuronal activity, a temporal window of 150 frames is used as input when using the CNN model. A detailed explanation of hyperparameter selection and models was provided in Supplementary Notes 1 and 2.

**Kalman filter**. In the position reconstruction experiment, a Kalman filter was used to reduce the decoding noise in the outputs of the decoder. The Kalman filter is one of the most widely used methods for position tracking. To estimate the state $\hat{x}_t$ [position and velocity] of the animal at time $t$, the estimation processes are defined as:

Time update:

$$\hat{x}_{\bar{t}} = \mathbf{A}\hat{x}_{t-1} + \mathbf{B}u_{t-1} \tag{4}$$

$$P_{\bar{t}} = \mathbf{A}P_{t-1}\mathbf{A}^T + \mathbf{Q} \tag{5}$$

Measurement update:

$$K_t = P_{\bar{t}}\mathbf{H}^T(\mathbf{H}P_{\bar{t}}\mathbf{H}^T + \mathbf{R})^{-1} \quad (6)$$

$$P_t = (\mathbf{I} - K_t\mathbf{H})P_{\bar{t}} \quad (7)$$

$$\hat{x}_t = \hat{x}_{\bar{t}} + K_t(z_t - \mathbf{H}\hat{x}_{\bar{t}}) \quad (8)$$

$$\mathbf{A} = \begin{bmatrix} 1 & \Delta t \\ 0 & 1 \end{bmatrix}, \mathbf{B} = \begin{bmatrix} \frac{1}{2}\Delta t^2 \\ \Delta t \end{bmatrix}, \mathbf{H} = [10] \quad (9)$$

where $\hat{x}_{\bar{t}}$ is the prior estimation of the state, $P_{\bar{t}}$ is the prior transition covariance, $K_t$ is the Kalman gain, $P_t$ is the updated transition covariance, $\hat{x}_t$ is the updated state, $u_{t-1}$ is the acceleration, $\mathbf{Q}$ is the transition covariance and $\mathbf{R}$ is the observation covariance. The sampling frequency is 30 fps, so $\Delta t$ equals 1/30. The initial values of the state were set to be zero and the transition covariance matrix was set to be identity. The values of $\mathbf{Q}$ and $\mathbf{R}$ were set to be 0.0001 and 1, respectively.

**Experimental procedures and analysis method**. All experiments consisted of two sessions: (1) a training session and (2) a real-time session. The training session was implemented offline to construct the decoding model, which was then used to decode neuronal activity in a real-time session.

**Real-time position reconstruction experiment**. Prior to the commencement of this experiment, mice were kept on dietary restriction and maintained at 85% of free-feeding body weights. The mice were trained to traverse a 1.6 m linear track for food rewards with miniscopes attached and were required to complete 12 trials of traversing each day for 1 week.

On the day of the training session, hippocampal neuronal activity was recorded when the mouse traversed a linear track for 12 trials. The sampling frequency of the miniscope was set to 30 fps and the animal's position was tracked using a video camera that was synchronized with the recording system. After recording, the translational frameshifting was corrected using a cross-correlation-based image registration algorithm[43]. The spatial footprints of neurons in the field of view were detected by implementing a constrained nonnegative matrix factorization for endoscopic recordings (CNMF-E) algorithm[44] with the centroid of each neuron and a small surrounding area used to measure activity. For each neuron, the raw calcium activity was defined as the average fluorescent intensity of the centroid pixel and the surrounding eight pixels. We then detected location-sensitive neurons using a shuffling method (see "Methods"). Next, signals from location-sensitive neurons were used to construct a decoding model to optimize position identification. The performance of the four decoders described above (GNB, SVM, MLP, and LSTM) was evaluated and compared. Finally, the decoding noise from the outputs of the decoder was reduced with a Kalman filter. The best decoder, together with the Kalman filter, was subsequently deployed in the real-time sessions.

In the real-time session, the mouse traversed the linear track for 12 trials. The position of the mouse was tracked using a video camera to assess the decoding accuracy. The same image registration method was implemented to align the image, and the raw calcium activity of position-sensitive neurons detected in the training session was extracted and fed into the decoder to reconstruct the animal's running trajectory in real time.

**Visual and auditory stimuli identification**. Hippocampal activity evoked by visual stimuli and auditory stimuli was studied separately. The mouse was put in a small opaque recording chamber and was exposed to either a flashlight or a speaker mounted above the chamber.

On the day of the training session, the mouse was placed in the chamber for 15 min before recording to acclimate. A Raspberry Pi 3 (Model B) board controlled a flashlight or a speaker. In the visual stimuli experiment, the flashlight was turned on for 2 s and off for 2 s alternately 150 times. In the auditory stimuli experiment, the speaker played three different frequency tones (4, 8, 16 kHz) randomly, 225 times with an activation period of 2 s and a mute period of 3 s. The data was analyzed using the same procedure described in the position reconstruction experiment: (1) image registration, (2) neuron centroid detection, (3) raw calcium activity extraction, (4) sensitive neuron detection, and (5) decoder construction. The performance of the GNB decoder, SVM decoder, and MLP neural network model were assessed and compared. In the auditory stimuli experiment, none of these three decoders showed outstanding performance in identifying the stimulus in each frame (see "Results"). A possible reason was that auditory information was encoded in temporal sequences, so we divided the data into several epochs for further analysis. The time length of the epoch was 5 s, which included the speaker activation and deactivation periods. Finally, a CNN model was constructed to decode the epoch data.

In the real-time session, the mouse was exposed to 150 light stimuli or 80 sound stimuli. The raw calcium activity of stimuli-sensitive neurons detected in the training session was extracted after image registration and provided to the decoder. The MLP model or SVM model that showed the best performance in the training session was deployed in visual stimuli experiments and a CNN model was deployed in auditory stimuli experiments to do the subsequent real-time decoding.

**Noise level and firing rate map similarity analysis**. The noise level was used to characterize the noise coupled to the calcium dynamics. Raw calcium activity was first processed with a zero-phase infinite impulse response lowpass filter (1 Hz cutoff frequency, filter order:20), and the signal noise level was defined as the difference between the raw calcium activity and the filtered activity.

To compare the consistency or the similarity of the firing rate map between the training session and the real-time session, normalized cross-correlation was measured[45]

$$R(u,v) = \frac{\sum_{xy}[m(x,y) - \bar{m}_{u,v}][n(x-u, y-v) - \bar{n}]}{\sqrt{\sum_{xy}[m(x,y) - \bar{m}_{u,v}]^2 \sum_{xy}[n(x-u, y-v) - \bar{n}]^2}} \quad (10)$$

where $m$ and $n$ represent the firing rate maps in the training session and the real-time session, $x$ and $y$ represent the pixels in the maps, and $u$ and $v$ represent the pixel shift along different dimensions in each map, respectively.

**Statistics and reproducibility**. The statistical analysis was conducted utilizing GraphPad PRISM version 7 software. Statistical significance was determined at $p$-value < 0.05. Statistical significance was determined through a two-tailed paired Student's $t$-test or one-way analysis of variance. The proposed analysis procedure was tested on three animals. Due to experimental design, the experimenter was not able to remain blinded to the manipulations carried out during the experiment.

**Reporting summary**. Further information on research design is available in the Nature Portfolio Reporting Summary linked to this article.

## Data availability

The data that support the findings of this study are available from the corresponding author upon reasonable request.

## Code availability

The code that supports the findings of this study is available from the corresponding author upon reasonable request.

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

## Acknowledgements

This work was supported by the Royal Melbourne Hospital Neuroscience Foundation (A2087) and the Australian Research Council under Discovery Project (DP170100363).

## Author contributions

D.S., R.R.U., and C.F. conceived and designed the study. D.S. carried out the experiment, processed the data, and drafted the manuscript. F.H. and Y.Y. processed the data. R.R.U. and C.F. did the critical manuscript revision. All authors have read and approved the final manuscript.

## Competing interests

The authors declare no competing interests.

## Ethical approval

Ethics approval was granted by the Florey Animal Ethics Committee (No. 18-008UM), subject to the restrictions contained in the Australian Code for the Care and Use of Animals for Scientific Purposes, 8th Edition.
