## [Peer Review File · Communications Engineering]

Reviewers' comments:

Reviewer #1 (Remarks to the Author):

In their manuscript, Sun and coworkers demonstrate that the raw intensity values from fluorescence brain imaging can be "interpreted" to understand physical movements and stimuli in a mouse model in real time.

In particular, their "optical brain-computer interface (OBCI)" has the apparent benefit of skipping preprocessing steps and thereby enabling faster analysis. The technique produces very nice accuracies; however, the tested conditions appear to be quite limited: a 1.6 m linear track was used.

In the discussion, the authors state: "We have decoded multiple sensory modalities from hippocampal neuronal activity in mice." I propose making the claim more precise to say that the method can recognize certain pre-trained patterns that are associated with specific behaviours. The distinction is important because an interpretation would seemingly allow for a flexible set of inputs, which I do not believe has been explored very much here - and may not have been a goal of the study.

I find the article is well-written and concise, making it generally accessible to a broader audience; nonetheless, given that this represents an important but restricted step toward using the transformation of calcium imaging for reconstructing physical movement paths, I believe it needs a broader range of demonstrations.

Lastly, given that one goal of the manuscript is a head-to-head comparison of different algorithms on the same data, I kindly suggest the authors make their data available openly rather than upon reasonable request. Given the large number of free-storage online repositories available, it would lower the barrier toward other groups that could test their interpretation algorithms on the data.

There is an opportunity to include in the supplementary material an expanded set of detailed explanations of experiments, as it currently consists of figures and tables only.

Reviewer #2 (Remarks to the Author):

In this manuscript, the authors performed 1-photon calcium imaging of CA1 hippocampus neurons and decoded spatial, visual, and auditory information from the neuronal ensemble activity in real-time using several types of decoder models. They compared the decoding accuracy and latency between GNB, SVM, MLP, LSTM, and CNN models.

Previous studies have already shown that the neurons in hippocampus encode spatial, visual, and auditory signals. Consistent with these previous studies, the authors successfully decoded these signals from CA1 neural activity. The novelty that the authors claim includes 1) the usage of calcium imaging for decoding, 2) the real-time application of the decoding, 3) decoding of non-spatial visual/auditory information. However, calcium imaging data has been widely used for spatial decoding from

hippocampus in previously published studies (e.g. Tu et al., *Neural Computation*, 2020; Murano et al., *PNAS*, 2022; Hazon et al., *Nature Communications*, 2022). Furthermore, short-latency decoding of spatial information for real-time application has been also reported previously with the positional error that seems to be even lower than this current study (Tu et al, *Neural Computation*, 2020). Given these previous studies, the main novelty of this study could be the real-time decoding of visual and auditory signals from hippocampus neurons. However, given that hippocampal coding of these information has been well known and the simplicity of those decoding tasks (categorical decoding), the scientific advance in this study is limited. The authors should revise the manuscript to tone down their novelty claims by acknowledging these previous studies. Furthermore, the authors should describe more details of their decoding analyses as the details are not sufficiently described in the current manuscript.

Other major points:

1. Although the authors claim that they succeeded in real-time decoding during animal behaviors, the total time it took from image acquisition to decoding is not shown. I see the processing time per frame for the decoding part in supplementary figures, but the time it took for the image data processing (e.g. image registration for motion correction, extraction of calcium signals) is not shown. For real-time decoding during animal behaviors, the processing time for image data also needs to be very short. Otherwise, the pipeline the authors propose is not suited for real-time application and brain-computer-interface.

2. It is unclear from the manuscript what kind of input was fed into each decoder. How many frames did they use as the inputs for each decoding analyses? Did authors use only the neural activity from the time points that are before the time point that they want to decode (e.g. If they want to decode the information at time t , did they use only the frames $t-1$, $t-2$,... but not $t+1$, $t+2$, ...)? If they used the activity from these future time points ($t+1$, $t+2$, ...), the decoding is not real-time in a strict sense.

3. The authors should describe the range and step of their grid search for hyperparameter selection for each decoder model. It is hard to gauge how much parameter space was explored for each type of decoder and whether the apparent performance differences between models could be due to insufficient optimization of each model type.

4. I suspect that the cross-correlation peaks at non-zero points for visual (and auditory) conditions could be due to sensory adaptation (Fig. S4c). My understanding is that the authors split the sessions into first 75% of data and the remaining 25% of data to calculate the cross-correlations between them. The sensory response in later trials would be affected by sensor adaptation, which may be the reason why the authors see non-zero peaks for vision condition. Do they see the same pattern even when they shift the time period of 25% from the last to the middle of a session? The authors should also clarify the unit of the x-axis of this plot (Is the time unit "frame"?).

5. For the auditory condition, the authors claim that only CNN worked well. However, it seems the authors used larger input frame numbers only for CNN, which makes difficult to assess whether the better accuracy is due to the use of CNN or larger input frame numbers. They should use consistent input data for the comparisons between different models.

6. The network architectures of selected networks models are unclear. The authors should include a table that lists the selected hyperparameter sets for each mouse and for each condition. It is also unclear what kind of CNN was used. For example, is it 1d CNN or 2d CNN? It is hard to interpret their results without knowing the details of their models.

7. Statistical tests are missing for many performance comparisons between models.

Minor points:

1. Lines 52-54 “This study presents, for the first time, a demonstration of hippocampal “cognitive decoding” using a novel multimodal real-time OBCI.” I do not think this study *demonstrates* cognitive decoding. Presence of spatial + visual + auditory signals is not equivalent to cognition.

2. Lines 222-224 “In contrast, our technique uses single-photon imaging, which in practice can detect many more neurons, allowing direct decoding of spatial, light, and sound modalities at high precision in real-time.” This sentence is misleading. 1p imaging gives poorer quality calcium data where the calcium signal from each cellular ROI is often contaminated with calcium signals from the other cells at out-of-focus range. In principle, 2p imaging allows higher quality calcium measurements. The number of neurons that can be imaged depends more on the density of GCaMP expressing cells. 2p imaging allows imaging from samples with denser GCaMP-expressing cells than with 1p imaging because of its higher spatial resolution.

Reviewers' comments:

Reviewer #1 (Remarks to the Author):

In their manuscript, Sun and coworkers demonstrate that the raw intensity values from fluorescence brain imaging can be "interpreted" to understand physical movements and stimuli in a mouse model in real time. In particular, their "optical brain-computer interface (OBCI)" has the apparent benefit of skipping preprocessing steps and thereby enabling faster analysis.

1. The technique produces very nice accuracies; however, the tested conditions appear to be quite limited: a 1.6 m linear track was used.

We would like to thank the reviewer for providing us with valuable feedback. This limitation is attributed to the confined space within the experimental room, with the issue of cable handling posing a more substantial challenge. A majority of researchers have opted for linear track lengths falling within the range of 1 to 2 meters. For instance, Lin et al. (2022) employed a 1-meter linear track, and Shuman et al. (2020) utilized a 2-meter linear track for their experiments. Managing longer cables can become challenging as the linear track extends. However, one solution to address the cable-related challenges is to use a wire-free miniscope (Barbera et al., 2019), which has recently become available on the market. We have included this information in the discussion section.

Lin, X., Chen, L., Baglietto-Vargas, D., Kamalipour, P., Ye, Q., LaFerla, F.M., Nitz, D.A., Holmes, T.C. and Xu, X., 2022. Spatial coding defects of hippocampal neural ensemble calcium activities in the triple-transgenic Alzheimer's disease mouse model. *Neurobiology of disease*, 162, p.105562.

Shuman, T., Aharoni, D., Cai, D.J., Lee, C.R., Chavlis, S., Page-Harley, L., Vetere, L.M., Feng, Y., Yang, C.Y., Mollinedo-Gajate, I. and Chen, L., 2020. Breakdown of spatial coding and interneuron synchronization in epileptic mice. *Nature neuroscience*, 23(2), pp.229-238.

Barbera, G., Liang, B., Zhang, L., Li, Y. and Lin, D.T., 2019. A wireless miniScope for deep brain imaging in freely moving mice. *Journal of neuroscience methods*, 323, pp.56-60.

2. In the discussion, the authors state: "We have decoded multiple sensory modalities from hippocampal neuronal activity in mice." I propose making the claim more precise to say that the method can recognize certain pre-trained patterns that are associated with specific behaviours. The distinction is important because an interpretation would seemingly allow for a flexible set of inputs, which I do not believe has been explored very much here - and may not have been a goal of the study.

Thank you for this suggestion. We have made this clear in the discussion section.

3. I find the article is well-written and concise, making it generally accessible to a broader audience; nonetheless, given that this represents an important but restricted step toward using the transformation of calcium imaging for reconstructing physical movement paths, I believe it needs a broader range of demonstrations.

In our upcoming work, we are actively working on further demonstrations and experiments to provide a more comprehensive validation of our real-time optical decoder driven by calcium activity. We trained a mouse to push a lever to get food reward and implanted a grin lens in the mouse motor cortex to record the neuronal calcium activity. Similar to our existing sensory modality decoder, we are in the process of training a model to decode the specific action of lever-pushing in real-time. Upon successful detection of this action by the decoder, a food reward will be dispensed. We hypothesize that once the animal receives the reward, it may no longer engage in lever-pushing to obtain additional rewards and instead may use its neural activity or mental commands to control. We have mentioned this future work in the manuscript.

4. Lastly, given that one goal of the manuscript is a head-to-head comparison of different algorithms on the same data, I kindly suggest the authors make their data available openly rather than upon reasonable request. Given the large number of free-storage online repositories available, it would lower the barrier toward other groups that could test their interpretation algorithms on the data.

Thank you for this suggestion, we are pleased to have uploaded the data to the cloud.

5. There is an opportunity to include in the supplementary material an expanded set of detailed explanations of experiments, as it currently consists of figures and tables only.

The experimental procedures and analysis methods are described in the Method section, specifically under 'Real-time position reconstruction experiment' and 'Visual and auditory stimuli identification.' Additionally, more detailed explanations regarding the decoding models used in the experiments have been included in the supplementary material.

Reviewer #2 (Remarks to the Author):

In this manuscript, the authors performed 1-photon calcium imaging of CA1 hippocampus neurons and decoded spatial, visual, and auditory information from the neuronal ensemble activity in real-time using several types of decoder models. They compared the decoding accuracy and latency between GNB, SVM, MLP, LSTM, and CNN models.

1. Previous studies have already shown that the neurons in hippocampus encode spatial, visual, and auditory signals. Consistent with these previous studies, the authors successfully decoded these signals from CA1 neural activity. The novelty that the authors claim includes 1) the usage of calcium imaging for decoding, 2) the real-time application of the decoding, 3) decoding of non-spatial visual/auditory information. However, calcium imaging data has been widely used for spatial decoding from hippocampus in previously published studies (e.g. Tu et al., Neural

Computation, 2020; Murano et al., PNAS, 2022; Hazon et al., Nature Communications, 2022). Furthermore, short-latency decoding of spatial information for real-time application has been also reported previously with the positional error that seems to be even lower than this current study (Tu et al, Neural Computation, 2020). Given these previous studies, the main novelty of this study could be the real-time decoding of visual and auditory signals from hippocampus neurons. However, given that hippocampal coding of these information has been well known and the simplicity of those decoding tasks (categorical decoding), the scientific advance in this study is limited. The authors should revise the manuscript to tone down their novelty claims by acknowledging these previous studies. Furthermore, the authors should describe more details of their decoding analyses as the details are not sufficiently described in the current manuscript.

We appreciate the valuable comments from the reviewer. Tu et al. (2020) developed an efficient strategy for decoding an animal's position using hippocampal calcium datasets from Dombeck et al. (2010) and Mau et al. (2018), and their maximum likelihood estimation method demonstrated good decoding accuracy. We have included this important reference in the manuscript. However, comparing the single-photon decoding accuracy between their study and ours presents challenges due to differences in experimental design. For instance, Tu et al. utilized a complex sensory-cued treadmill, while we employed a more straightforward linear track. Furthermore, the animal had to maintain a stable running speed on the treadmill, and running speed can impact place cell activity (Geisler, 2007). We also employed a higher sampling frequency of 30 fps in contrast to their use of 20 fps, and the sampling frequency is known to affect the calcium signal quality. Moreover, our training methods differed, making it challenging to ascertain whether the animals were trained to the same level.

The reviewer also mentioned the work from Hazon and Murano. Hazon et al. (2022) used an SVM decoder for spatial decoding, but this method requires lots of computational processing and is not implemented in real-time. Murano et al. (2022) observed neuronal activity in the dentate gyrus, while we were looking at the hippocampal CA1 region.

While both Tu et al. and our study used different decoders for position reconstruction, a noteworthy innovation in our work is the integration of a Kalman filter after neural network decoding to reduce inherent decoding noise. Regarding non-spatial stimuli, previous research has shown that non-spatial factors can modulate the activity of hippocampal neurons, including auditory and visual stimuli. However, these non-spatial responses have been found to be spatial context-dependent or task-dependent in almost all previous studies (see O'Keefe and Krupic, 2021 for review). Consequently, such responses may actually represent the location or context in which the feature is present rather than the feature per se. It is still not entirely clear whether non-spatial responses can be reliably decoded in the absence of accompanying spatial input. This uncertainty prompted us to place the animal in a small recording chamber, eliminating spatial input, in order to investigate whether non-spatial information could be decoded under these conditions. Another minor innovation lies in the redesign of the miniscope optical system, which has resulted in a larger field of view. We have revised the manuscript to provide clarity on these points and to acknowledge the previous studies mentioned by the reviewer. Additionally, we have included more details on decoding analyses in the manuscript.

Geisler, C., Robbe, D., Zugaro, M., Sirota, A. and Buzsáki, G., 2007. Hippocampal place cell assemblies are speed-controlled oscillators. *Proceedings of the National Academy of Sciences*, 104(19), pp.8149-8154.

O'Keefe, J. and Krupic, J., 2021. Do hippocampal pyramidal cells respond to nonspatial stimuli?. *Physiological Reviews*, 101(3), pp.1427-1456.

2. Although the authors claim that they succeeded in real-time decoding during animal behaviors, the total time it took from image acquisition to decoding is not shown. I see the processing time per frame for the decoding part in supplementary figures, but the time it took for the image data processing (e.g. image registration for motion correction, extraction of calcium signals) is not shown. For real-time decoding during animal behaviors, the processing time for image data also needs to be very short. Otherwise, the pipeline the authors propose is not suited for real-time application and brain-computer-interface.

At a frame rate of 30Hz, the frame interval is approximately 33ms. To achieve consistent frame-by-frame decoding, the data processing time should be kept below 33ms to prevent the camera data from overflowing the PC buffer. The footprints of neurons are detected offline, and the extraction of raw calcium signals only requires simple matrix operation. The overall processing time, including image registration and calcium signal extraction, is approximately 2.2 ms. The decoding time varies depending on the model but is less than 3ms. Consequently, the total time consumption is less than 5.2ms per frame. We have included these details in the manuscript.

3. It is unclear from the manuscript what kind of input was fed into each decoder. How many frames did they use as the inputs for each decoding analyses? Did authors use only the neural activity from the time points that are before the time point that they want to decode (e.g. If they want to decode the information at time t , did they use only the frames $t-1, t-2, \dots$ but not $t+1, t+2, \dots$)? If they used the activity from these future time points ($t+1, t+2, \dots$), the decoding is not real-time in a strict sense.

Apologies for any potential confusion. We utilized only the neural activity recorded before the time point of interest for decoding, so the decoding is in real-time. In our decoding methods, such as the Gaussian Naïve Bayes decoder, Support Vector Machine decoder, and Multilayer Perceptron Neural Network decoder, we used single frame data. However, when employing the Long Short-Term Memory (LSTM) neural network model, we tested different window lengths and determined that a temporal window of 5 frames typically provided accurate decoding. As the auditory stimuli have a long temporal effect on neuronal activity, a temporal window of 150 frames is used as input when using the CNN model. These specific details have been included in the Methods section of the manuscript for clarity.

4. The authors should describe the range and step of their grid search for hyperparameter selection for each decoder model. It is hard to gauge how much parameter space was explored for each type of decoder and whether the apparent performance differences between models could be due to insufficient optimization of each model type.

We agree describing the step size is useful in this case. In this paper, our primary objective is not solely to determine the most accurate decoder, but rather to establish a functional model. Given the substantial workload involved in testing multiple models using datasets from three animals and three different experiments, we opted not to conduct a comprehensive grid search for hyperparameter optimization in certain cases. Instead of simultaneously optimizing all hyperparameters, we adopted a strategic approach. We began by identifying and tuning the most impactful hyperparameters while keeping others at reasonable values. The testing values for the grid search were determined based on previous experience and some random testing. Our hyperparameter optimization process varied for different models. For the MLP model, we followed a sequential approach. We initially set the number of epochs to 50 and the unit numbers for hidden layers 1 and 2 to 150 and 30, respectively. Subsequently, we determined the optimal learning rate while maintaining fixed values for other hyperparameters. Once the learning rate was determined, we proceeded to optimize the unit numbers for hidden layers 1 and 2, followed by fine-tuning the number of epochs. For the LSTM model, we optimized hyperparameters in the following order: sliding time window size, learning rate, units for layers 1 and 2 in LSTM, and epoch number. For the CNN model, our optimization order included learning rate, filter numbers, kernel size, pool size, and epoch number. We have incorporated these specific details into the manuscript.

SVM model:

C {0.01, 0.1, 1, 10, 100}

MLP model:

learning rate {0.0001, 0.001, 0.01, 0.1}

hidden layer 1 units {150, 200, 250, 300}

hidden layer 2 units {30, 50, 150, 200}

epoch number {50, 100, 150, 200}

LSTM model:

learning rate {0.0001, 0.001, 0.01, 0.1}

Layer 1 LSTM units {32, 64, 128}

Layer 2 LSTM units {32, 64, 128}

Sliding time window size {5, 10, 15, 20}

epoch number {50, 75, 100, 150}

CNN model:

learning rate {0.0001, 0.001, 0.01, 0.1}

CNN layer 1 filters {16, 32, 64}

CNN layer 1 pool_size {{2, 1}, (3, 1)}

CNN layer 1 kernel_size {{3, 1}, {5,1}}

CNN layer 2 filters {16, 32, 64}

CNN layer 2 pool_size {{2, 1}, (3, 1)}

CNN layer 2 kernel_size {{3, 1}, {5,1}}

epoch number {50, 100, 150, 200}

5. I suspect that the cross-correlation peaks at non-zero points for visual (and auditory) conditions could be due to sensory adaptation (Fig. S4c). My understanding is that the authors split the sessions into first 75% of data and the remaining 25% of data to calculate the cross-correlations between them. The sensory response in later trials would be affected by sensor adaptation, which may be the reason why the authors see non-zero peaks for vision condition. Do they see the same pattern even when they shift the time period of 25% from the last to the middle of a session? The authors should also clarify the unit of the x-axis of this plot (Is the time unit “frame”?).

We agree that the cross-correlation peaks at non-zero points for visual and auditory conditions could be attributed to sensory adaptation. The cross-correlation measured the similarity of the neuronal firing rate maps between the training session and the real-time session, without utilizing data divisions such as 75% and 25% within a single session. It's important to note that this method primarily captures the fundamental characteristics of population activity and cannot discern “deep features” within the neuronal firing pattern. Considering that the neural network decoding model was able to provide accurate decoding even in this scenario, it may imply that neuronal activity related to nonspatial stimuli is better characterized by deeper features. This could potentially explain why some previous studies have reported a low sensitivity of hippocampal neurons to these non-spatial stimuli (O’Keefe and Krupic, 2021), possibly due to the analysis method. The time unit is frame (the miniscope recording frequency is 30 frames/s), and the unit of the x-axis of the plot is the number of frames. We have included all these in the manuscript.

O’Keefe, J. and Krupic, J., 2021. Do hippocampal pyramidal cells respond to nonspatial stimuli?. *Physiological Reviews*, 101(3), pp.1427-1456.

6. For the auditory condition, the authors claim that only CNN worked well. However, it seems the authors used larger input frame numbers only for CNN, which makes difficult to assess whether the better accuracy is due to the use of CNN or larger input frame numbers. They should use consistent input data for the comparisons between different models.

We agree that it is not appropriate to directly compare the decoding accuracy between using Convolutional Neural Networks (CNN) and other models, due to differences in the number of input data. Initially, we tested the performance of GNB, SVM, and MLP decoders using single-frame data as input. However, the decoding accuracy was extremely low, likely due to the long-time effects of auditory stimuli on neuronal activity. Then we decided to increase the number of input frames to better capture the temporal aspects of the data. Nevertheless, GNB, SVM, and MLP models are less effective in handling temporal multi-dimensional data. A common approach to analyze such data is to flatten or reshape the inputs. However, this method significantly increases dimensionality, which brings about challenges such as “the curse of dimensionality”, heightened computational complexity, and the potential loss of spatial information in the features. Given these challenges, along with substantial optimization time consumption, we chose not to use these models. In contrast, CNN has been widely employed for the analysis of temporal multidimensional data, making it a more suitable choice for our data analysis. We have provided clarification on this point in the manuscript.

7. The network architectures of selected networks models are unclear. The authors should include a table that lists the selected hyperparameter sets for each mouse and for each condition. It is also unclear what kind of CNN was used. For example, is it 1d CNN or 2d CNN? It is hard to interpret their results without knowing the details of their models.

We have included the network structure and all hyperparameters in the manuscript.

Linear track:

Mouse 1: MLP

Dense layer_1 (nodes= 250, activation='relu', kernel_initializer='random_normal')

Dense layer_2 (nodes= 150, activation='relu', kernel_initializer='random_normal')

Dense layer_3 (nodes= 80, activation='softmax', kernel_initializer='random_normal')

Model (optimizer='Adam', learning_rate=0.001, loss='categorical_crossentropy', metrics=['accuracy'], epochs=100, batch_size=32)

Mouse 2: MLP

Dense layer_1 (nodes= 250, activation='relu', kernel_initializer='random_normal')

Dense layer_2 (nodes= 150, activation='relu', kernel_initializer='random_normal')

Dense layer_3 (nodes= 80, activation='softmax', kernel_initializer='random_normal')

Model (optimizer='Adam', learning_rate=0.001, loss='categorical_crossentropy', metrics=['accuracy'], epochs=100, batch_size=32)

Mouse 3: SVM

SVC(C=10, gamma=1/N), where N is the reciprocal of the number of input features.

Visual stimuli:

Mouse 1: SVM

SVC(C=1, gamma=1/N)), where N is the reciprocal of the number of input features.

Mouse 2: SVM

SVC(C=1, gamma=1/N)), where N is the reciprocal of the number of input features.

Mouse 3: MLP

Dense layer_1 (nodes= 250, activation='relu', kernel_initializer='random_normal')

Dense layer_2 (nodes= 150, activation='relu', kernel_initializer='random_normal')

Dense layer_3 (nodes= 80, activation='softmax', kernel_initializer='random_normal')

Model (optimizer='Adam', learning_rate=0.001, loss='categorical_crossentropy',
metrics=['accuracy'], epochs=100, batch_size=32)

Auditory stimuli:

Mouse 1: CNN

Input(shape = input_shape)

Conv2D(filters = 32, kernel_size=(3, 1), activation="relu", padding='same')

MaxPooling2D (pool_size=(2, 1))

Conv2D(filters = 32, kernel_size=(3, 1), activation="relu", padding='same')

MaxPooling2D (pool_size=(2, 1))

Flatten()

Dropout(0.2)

Dense(3, activation="softmax")

Model (optimizer='Adam', learning_rate=0.001, loss='categorical_crossentropy',
metrics=['accuracy'], epochs=100, batch_size=32)

Mouse 2: CNN

Input(shape = input_shape)

Conv2D(filters = 64, kernel_size=(3, 1), activation="relu", padding='same')

MaxPooling2D (pool_size=(2, 1))

Conv2D(filters = 64, kernel_size=(3, 1), activation="relu", padding='same')

MaxPooling2D (pool_size=(2, 1))

Flatten()

Dropout(0.2)

Dense(3, activation="softmax")

Model (optimizer='Adam', learning_rate=0.001, loss='categorical_crossentropy',
metrics=['accuracy'], epochs=150, batch_size=32)

Mouse 3: CNN

Input(shape = input_shape)

Conv2D(filters = 32, kernel_size=(5, 1), activation="relu", padding='same')

MaxPooling2D (pool_size=(3, 1))

Conv2D(filters = 64, kernel_size=(5, 1), activation="relu", padding='same')

MaxPooling2D (pool_size=(3, 1))

Flatten()

Dropout(0.2)

Dense(3, activation="softmax")

Model (optimizer='Adam', learning_rate=0.001, loss='categorical_crossentropy',
metrics=['accuracy'], epochs=100, batch_size=32)

8. Statistical tests are missing for many performance comparisons between models.

We have added statistical test results for performance comparisons between models in the manuscript.

Minor points:

1. Lines 52-54 “This study presents, for the first time, a demonstration of hippocampal “cognitive decoding” using a novel multimodal real-time OBCI.” I do not think this study *demonstrates* cognitive decoding. Presence of spatial + visual + auditory signals is not equivalent to cognition.

We have changed the words to “multisensory modality decoding”.

2. Lines 222-224 “In contrast, our technique uses single-photon imaging, which in practice can detect many more neurons, allowing direct decoding of spatial, light, and sound modalities at high precision in real-time.” This sentence is misleading. 1p imaging gives poorer quality calcium data where the calcium signal from each cellular ROI is often contaminated with calcium signals from the other cells at out-of-focus range. In principle, 2p imaging allows higher quality calcium measurements. The number of neurons that can be imaged depends more on the density of GCaMP expressing cells. 2p imaging allows imaging from samples with denser GCaMP-expressing cells than with 1p imaging because of its higher spatial resolution.

Apologies for any potential confusion. We agree that one-photon imaging yields poorer signal

quality compared to two-photon imaging since it collects neuronal firing activity of out-of-focus cells as well. However, it's important to note that this additional neuronal firing information has the potential to enhance decoding accuracy. We have provided clarification on this in the manuscript."

REVIEWERS' COMMENTS:

Reviewer #1 (Remarks to the Author):

I thank the authors for their detailed and careful corrections, particularly clarifying and qualifying the specific advances in their publication as well as their added commentary on future studies that will continue to expand on their work.

I believe the manuscript is ready for publication.

Reviewer #2 (Remarks to the Author):

I thank the authors for their responses. In their revised manuscript, the authors have added relevant citations, included the details of their models, hyperparameter searches, and latency information for their real-time application. Overall, I think these additions have resulted in a much improved manuscript. Although I feel the model selections in this study are somewhat arbitrary and not thorough, I think the added model details in this study are valuable resources for others to design decoders for real-time applications. One minor point: p-values between CNN and the other models are not shown in the supplementary fig3 legend.

REVIEWERS' COMMENTS:

Reviewer #1:

Remarks to the Author:

I thank the authors for their detailed and careful corrections, particularly clarifying and qualifying the specific advances in their publication as well as their added commentary on future studies that will continue to expand on their work.

I believe the manuscript is ready for publication.

We thank the reviewer for the time and effort in reading our revised manuscript and providing feedback.

Reviewer #2:

Remarks to the Author:

I thank the authors for their responses. In their revised manuscript, the authors have added relevant citations, included the details of their models, hyperparameter searches, and latency information for their real-time application. Overall, I think these additions have resulted in a much improved manuscript. Although I feel the model selections in this study are somewhat arbitrary and not thorough, I think the added model details in this study are valuable resources for others to design decoders for real-time applications. One minor point: p-values between CNN and the other models are not shown in the supplementary fig3 legend.

We thank the reviewer for providing us with the feedback. We have added p-values between CNN and the other models in the supplementary material.